# Multiple neurons encode CrebB dependent appetitive long-term memory in the mushroom body circuit

Yves F Widmer, Cornelia Fritsch, Magali M Jungo, Silvia Almeida, Boris Egger, Simon G Sprecher*

Department of Biology, University of Fribourg , Fribourg, Switzerland

**Abstract** Lasting changes in gene expression are critical for the formation of long-term memories (LTMs), depending on the conserved CrebB transcriptional activator. While requirement of distinct neurons in defined circuits for different learning and memory phases have been studied in detail, only little is known regarding the gene regulatory changes that occur within these neurons. We here use the fruit fly as powerful model system to study the neural circuits of CrebB-dependent appetitive olfactory LTM. We edited the *CrebB* locus to create a GFP-tagged *CrebB* conditional knockout allele, allowing us to generate mutant, post-mitotic neurons with high spatial and temporal precision. Investigating CrebB-dependence within the mushroom body (MB) circuit we show that MB $\alpha/\beta$ and $\alpha'/\beta'$ neurons as well as MBON $\alpha3$, but not in dopaminergic neurons require CrebB for LTM. Thus, transcriptional memory traces occur in different neurons within the same neural circuit.
DOI: https://doi.org/10.7554/eLife.39196.001

## Introduction

Short-term memories (STMs) rely molecularly on rapidly acting biochemical processes altering the weight of defined synaptic connections. In contrast, long-lasting forms of memories require more permanent molecular changes including alterations of the transcriptional program. The cAMP response element binding protein (CREB) is an evolutionarily conserved basic leucine zipper transcription factor, which is involved in long-term memory (LTM). Activated CREB binds to cAMP response element (CRE) sites of target gene regulatory regions promoting transcription. CREB is regulated by numerous cellular events, responding to a diverse set of physiological stimuli. It has been shown that the transcription factor CREB is implicated in many biological processes, including circadian rhythm, stress, metabolism, cell survival and drug adaptation (*Mayr and Montminy, 2001*; *Carlezon et al., 2005*). However, CREB is best known for its role in memory formation. Studies in vertebrates and invertebrates showed that CREB-dependent gene transcription is a crucial component of long-term memory formation, but not of short-term memory (*Yin and Tully, 1996*; *Silva et al., 1998*; *Kida and Serita, 2014*). The *Drosophila* genome contains two CREB-like genes: *CrebA* and *CrebB* (*Smolik et al., 1992*; *Usui et al., 1993*). *CrebB*, that produces multiple protein isoforms, has been shown to be essential for LTM in flies (*Yin et al., 1994*).

Fruit flies associate odors with different reinforcing stimuli. Sugar as reward paired with an odor, induces LTM that requires protein synthesis and lasts for several days. Flies display memory as a selective approach of the previously sugar-paired odor (*Krashes and Waddell, 2008*; *Colomb et al., 2009*). The mushroom body (MB) is regarded as a main center of olfactory associative memory in the *Drosophila* brain (*Heisenberg et al., 1985*; *de Belle and Heisenberg, 1994*; *Dubnau et al., 2001*). MB intrinsic neurons forming the predominant lobe system are called Kenyon cells (KCs). Olfactory information is transmitted by projection neurons (PNs) from the antennal lobe to the calyx

*For correspondence: simon.sprecher@unifr.ch

**Competing interests:** The authors declare that no competing interests exist.

**eLife digest** Our brains can store different types of memories. You may have forgotten what you had for lunch yesterday, but still be able to remember a song from your childhood. Short-term memories and long-term memories form via different mechanisms. To establish long-term memories, the brain must produce new proteins, many of which are common to all members of the animal kingdom. By studying these proteins in organisms such as fruit flies, we can learn more about their role in our own memories.

Widmer et al. used this approach to explore how a protein called CrebB helps fruit flies to remember for several days that a specific odor is associated with a sugary reward. These odor-reward memories form in a brain region called the mushroom body, which has three lobes. Input neurons supply information about the odor and the reward to the region, while output neurons pass on information to other parts of the fly brain. CrebB regulates the production of new proteins required to form these long-term odor-reward memories: but where exactly does CrebB act during this process?

Using a gene editing technique called CRISPR, Widmer et al. generated mutant flies. In these insects CrebB could be easily deactivated 'at will' in either the entire brain, the whole mushroom body, each of the three lobes or in specific output neurons. The flies were then trained on the odor-reward task, and their memory tested 24 hours later. The results revealed that for the memories to form, CrebB is only required in two of the three lobes of the mushroom body, and in certain output neurons. Future studies can now focus on the cells shown to need CrebB to create long-term memories, and identify the other proteins involved in this process.

In humans, defects in CrebB are associated with intellectual disability, addiction and depression. The mutant fly created by Widmer et al. could be a useful model in which to investigate how the protein may play a role in these conditions.
DOI: https://doi.org/10.7554/eLife.39196.002

of the MB, where they directly synapse onto KCs. KCs can be further subdivided into three major morphological and functional types, α/β, α′/β′ and γ neurons, forming the lobe system of the MB (*Crittenden et al., 1998*). MB lobes are tiled by spatially separated input from Dopaminergic neurons (DANs) as well as dendritic arbors of MB output neurons (MBONs). Dopaminergic innervation provides critical input to the MB circuit as conditioning signal. DANs of the protocerebral anterior medial (PAM) cluster respond to sugar and convey the reinforcing effect of sugar to defined MB lobe compartments, while other DANs are involved in other types of memories (*Burke et al., 2012*; *Liu et al., 2012*). Output of the KCs is transmitted to other brain regions by 34 MBONs that fall into 21 discrete anatomical classes (*Aso et al., 2014*).

While there is a wide consensus that CrebB is involved in LTM formation, it remains less clear in which neurons CrebB is required to induce LTM. Previous studies using overexpression of a *CrebB* repressor or RNAi to inhibit CrebB in the MB and MBONs resulted in conflicting interpretation of CrebB requirement (*Yin et al., 1994*; *Yu et al., 2006*; *Chen et al., 2012*; *Hirano et al., 2013*). To shed light onto this question we created a *CrebB* conditional knockout allele ($CrebB^{cKO}$) allowing us to study CrebB function in defined cells. Using the CRISPR/Cas9 system, we replaced the endogenous *CrebB* gene with a GFP-tagged *CrebB* gene flanked by FRT sites. The generation of an in-frame CrebB::GFP fusion protein allowed us to precisely monitor CrebB expression and localization in the nervous system. Moreover, flippase (FLP) mediated FRT recombination allowed us to delete *CrebB::GFP* from the genome in genetically accessible cells hereby generating null mutant cells, within an otherwise wildtype animal. We used cell-type specific CrebB knockout in defined sets of neurons of the mushroom body circuit and assessed appetitive olfactory long-term memory. Our findings show that removing CrebB from the MB results in animals that are able to form STM, but no LTM, indeed highlighting the critical function of CrebB for LTM in the MB. Interestingly, removing CrebB in α/β and α′/β′, but not in the γ lobe affected LTM further supporting the differential role of the three major MB lobe system in different phases of memory formation. We found that removing CrebB from the dopaminergic PAM neurons did not affect LTM, while genetic excision of the CrebB

gene in MBON α3 severely reduced LTM formation. Thus, multiple neurons require CrebB within the same neural circuit to form an appetitive LTM trace.

## Results

### Generation and verification of a *CrebB* conditional knockout allele

To investigate the spatial requirement of CrebB for LTM formation within the mushroom body circuit we generated a *CrebB* conditional knockout allele (*CrebB^{cKO}*) using the CRISPR/Cas9 system (*Bassett and Liu, 2014*). In short, two FRT sites and an N-terminal *eGFP* tag were introduced into the *CrebB* locus. The first CRISPR site was selected immediately upstream of the *CrebB* start codon and the second used CRISPR site was located before the last exon of *CrebB* thus replacing the coding region of the locus with a donor template sequence for homology-directed repair (HDR) containing two FRT sites flanking the coding sequence of *eGFP* and the *CrebB* genomic sequence (*Figure 1A,B*). The successful generation of the *CrebB^{cKO}* allele was confirmed by sequencing of the 3' and 5' introduced GFP and FRT sites. The FRT flanked *CrebB::GFP* locus allows visualization of CrebB protein localization and proof of flippase recombinase mediated removal of the fusion protein by the absence of GFP expression. Using different *Gal4* driver lines and *Gal80^{ts}*, a temperature sensitive inhibitor of *Gal4*, spatially and temporally controlled *UAS-FLP* expression was performed (*McGuire et al., 2003*). In FLP expressing cells, GFP-tagged *CrebB* is deleted from the genome, creating *CrebB* mutant cells (*Figure 1B*).

We first assessed the engineered FRT- *GFP::CrebB* -FRT locus for proper CrebB protein expression by generating a guinea-pig polyclonal anti-CrebB antibody raised against the CrebB full-length protein sequence of isoform F. CrebB::GFP is expressed in virtually all cells of the brains of *CrebB^{cKO}* flies. Co-localization of anti-CrebB and anti-GFP antibody shows successful tagging of CrebB with GFP and further supports the pan-neuronal expression of CrebB (*Figure 1C* and *Figure 1—figure supplement 1*). *UAS-FLP; nSyb-Gal4* was used to remove CrebB::GFP in all neurons. To prevent developmental defects, we temporally restricted FLP expression using a *tubGal80^{ts}* transgene (*McGuire et al., 2003*). Knockout of CrebB was induced after hatching by moving animals to 29°C for 6 days before antibody staining or conditioning experiments. Brains from flies with induced pan-neuronal CrebB deletion were dissected and antibody staining performed. Expression of CrebB::GFP in the brain was completely lost in neurons and only detectable in Repo-positive glia cells confirming the efficiency of the created *CrebB* conditional knockout (*Campbell et al., 1994*; *Xiong et al., 1994*; *Halter et al., 1995*) (*Figure 2A,B*).

### CrebB is required for LTM, but dispensable for STM and MTM

To study the role of CrebB in memory formation, we used a classical olfactory conditioning paradigm, in which flies learned to approach an odor that was previously associated with sugar reward (*Krashes and Waddell, 2008*; *Colomb et al., 2009*). We tested flies directly after conditioning to measure short-term memory (STM), after 3 h to measure middle-term memory (MTM) or after 24 h to measure long-term memory (LTM). We first tested flies with induced pan-neuronal CrebB knockout along with the corresponding parental lines for STM. Since *CrebB* is located on the X chromosome, we always calculated memory performance indices of male and female animals separately. Male offspring flies of *CrebB^{cKO}; tubGal80^{ts}* females crossed with *+;UAS-FLP; nSyb-Gal4* males have only the *CrebB^{cKO}* allele (*CrebB^{cKO}/Y*) and therefore are *CrebB* null mutants after flippase induced excision. Female progeny of the cross have in addition to *CrebB^{cKO}* a wild type allele of *CrebB* and served as control group together with male flies of the parental lines. The tested groups did not show significantly different memory scores measured immediately after training, confirming dispensability of CrebB for STM (*Figure 2C*). We next assessed MTM and found that also for this memory phase CrebB is not required in neurons (*Figure 2D*). To evaluate LTM, flies were tested 24 h after conditioning. Unflipped male *CrebB^{cKO}/Y* flies showed intact LTM and did not perform significantly different from female *CrebB^{cKO}/+*, which further confirmed that conducted genome editing in the CrebB locus does not interfere with the function of CrebB in LTM formation. However, males with induced CrebB knockout in all neurons showed drastically reduced 24 h LTM performance (*Figure 2E*). The memory performance of non-induced flies kept continuously at 18°C was similar to that of genetic controls (*Figure 2—figure supplement 1*).

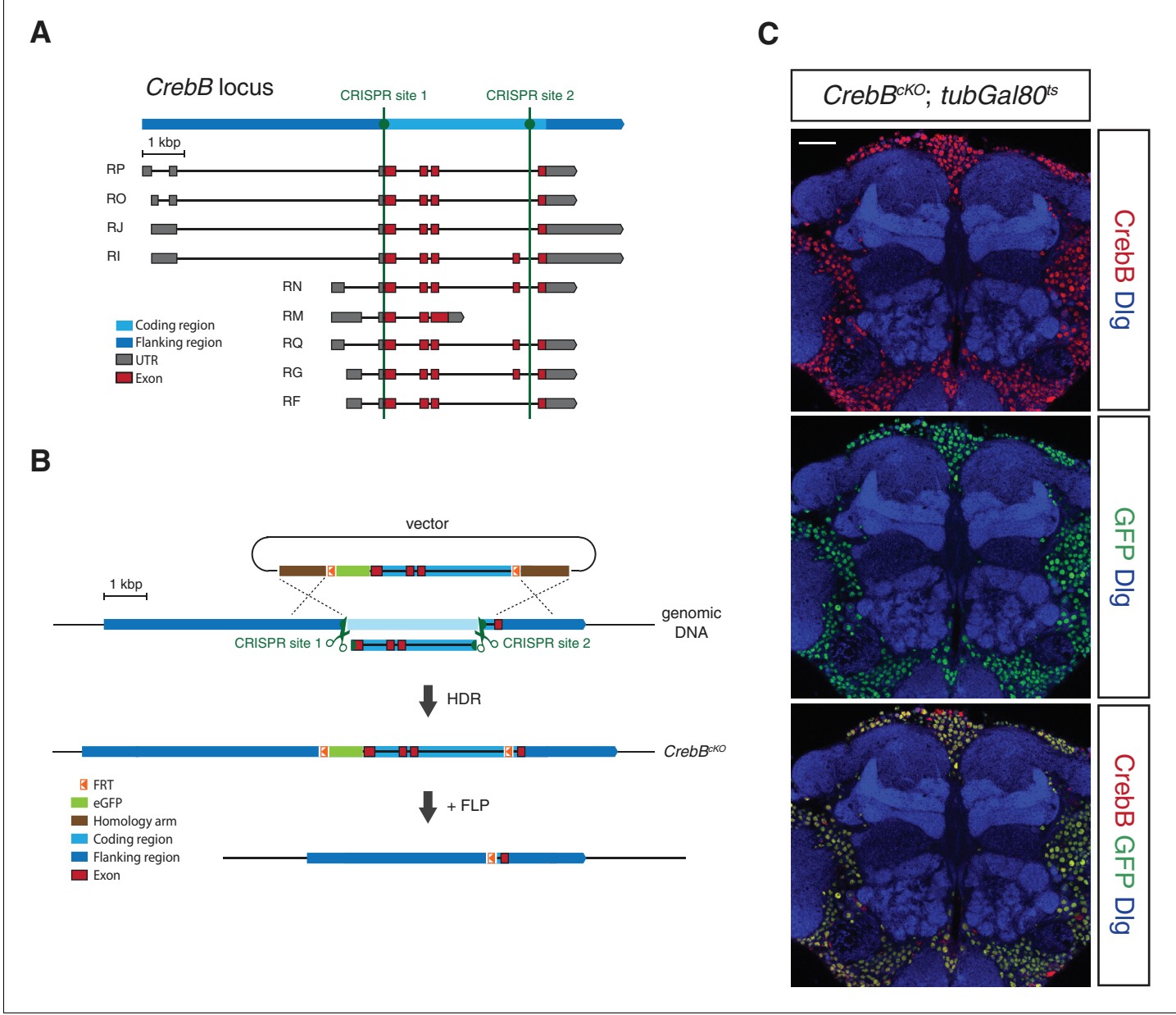

**Figure 1.** Generation of a conditional knockout allele for *CrebB*. (**A**) Schematic representation of the *CrebB* gene locus with nine different transcript isoforms. The location of the two used CRISPR sites are highlighted in green. (**B**) The *CrebB* conditional knockout allele (*CrebB^cKO^*) was generated using the CRISPR/Cas9 technique. The donor vector contained the coding region of *CrebB* with *eGFP* in front and two FRT sites flanking this sequence. The endogenous *CrebB* coding region was replaced by the donor DNA through CRISPR/Cas9 mediated homology-directed repair (HDR). In the resulting *CrebB^cKO^* allele the inserted *eGFP* and *CrebB* coding sequence can be removed by flippase (FLP) recombinase. (**C**) CrebB::GFP expression in a frontal brain confocal section of *CrebB^cKO^; tubGal80^ts^* flies was visualized using anti-GFP (green) and anti-CrebB (red) antibodies. Brain structures were labeled with anti-Discs large (Dlg, blue) antibodies. Scale bar: 30 μm.

DOI: https://doi.org/10.7554/eLife.39196.003

The following figure supplement is available for figure 1:

**Figure supplement 1.** Expression pattern of CrebB::GFP.
DOI: https://doi.org/10.7554/eLife.39196.004

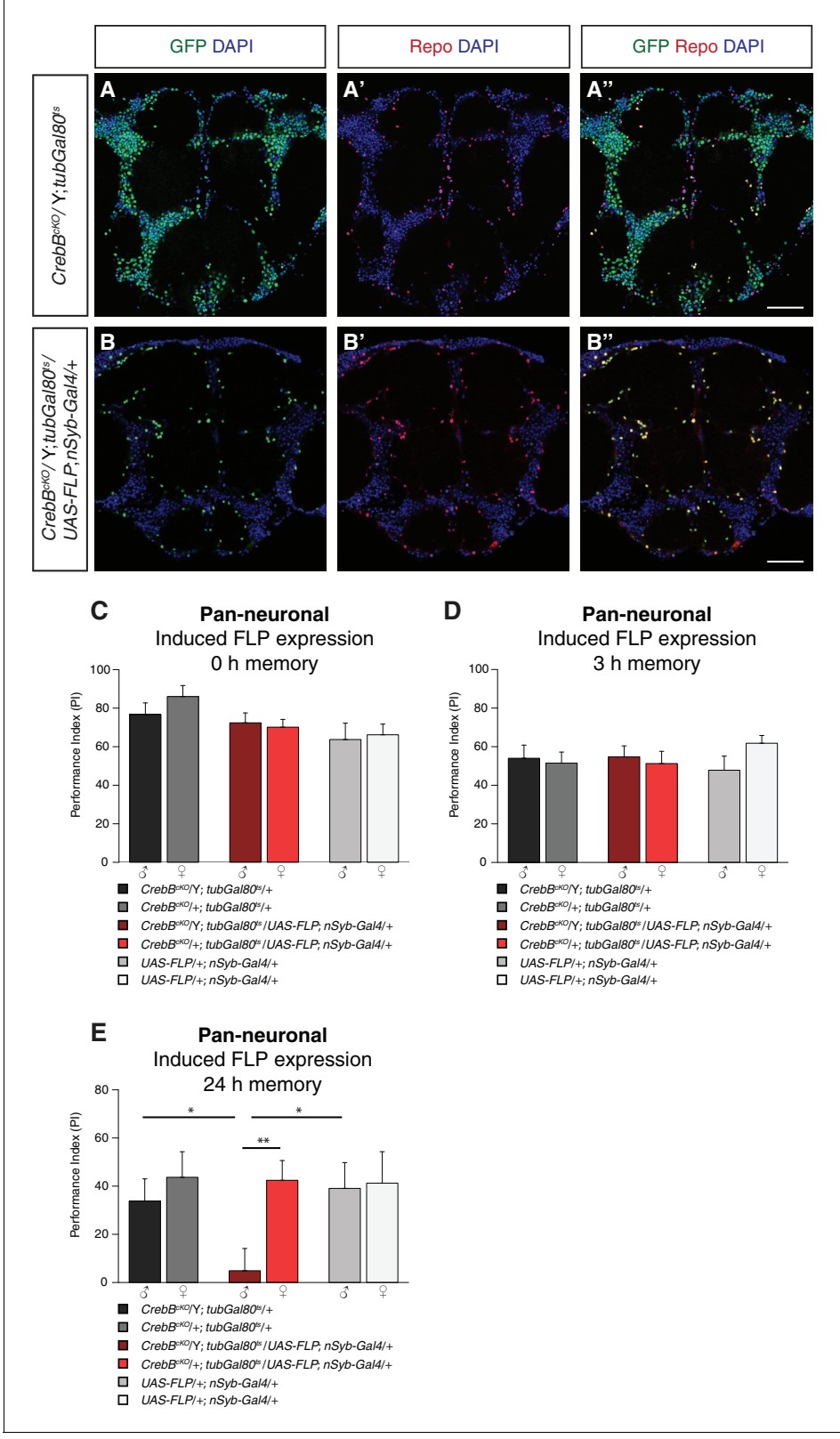

**Figure 2.** CrebB is dispensable for STM and MTM, but required for LTM. *CrebB::GFP* was removed from neurons and antibody stainings were conducted along with appetitive olfactory conditioning experiments. (**A, B**) Brains were stained with anti-GFP antibodies (green), anti-Repo antibodies (red) and DAPI (blue). (**A, A', A'**) In *CrebB^cKO*, *tubGal80^ts* brains, CrebB::GFP was detected in most of the cells. (**B, B', B''**) After pan-neuronal induction of CrebB::GFP knockout, GFP was only expressed in Repo-positive glia cells. Scale bars: 50 μm. (**C–E**) Short-term, middle-term and long-term memory

*Figure 2 continued on next page*

*Figure 2 continued*
performance was assessed. (C,D) Pan-neuronal CrebB knockout did not affect STM (N $\geq$ 10) and MTM (N = 8). (E) LTM, tested 24 h after conditioning, was severely impaired in flies with induced CrebB knockout in all neurons. Memory performance of male *CrebB$^{cKO}$/Y; tubGal80$^{ts}$/UAS-FLP; nSyb-Gal4/+* flies was significantly lower than in female flies (*CrebB$^{cKO}$/+; tubGal80$^{ts}$/UAS-FLP; nSyb-Gal4/+*), and in male flies of the parental control flies. N $\geq$ 9. Bar graphs represent the mean and error bars represent the standard error of the mean (SEM). Asterisks denote significant differences between groups; *p<0.05, **p<0.01 (Welch two sample *t*-test).
DOI: https://doi.org/10.7554/eLife.39196.005
The following figure supplement is available for figure 2:

**Figure supplement 1.** LTM is not impaired in non-induced CrebB knockout flies.
DOI: https://doi.org/10.7554/eLife.39196.006

## CrebB knockout in α/β and α′/β′, but not γ neurons impairs LTM

We next tested the requirement of CrebB in the mushroom body, a central structure for olfactory associative memory (*Heisenberg et al., 1985*; *de Belle and Heisenberg, 1994*; *Dubnau et al., 2001*; *Crittenden et al., 1998*). To remove CrebB from the entire MB we used the *OK107-Gal4* line, which is expressed in most of the KCs across the different lobes (*Aso et al., 2009*). To confirm the CrebB removal we stained with the KC marker Eyeless (Ey), which showed complete absence of CrebB::GFP from the entire MB six days after knockout induction (*Kurusu et al., 2000*) (*Figure 3B*). Next, we tested MB-specific CrebB knockout flies for their memory performance. While control animals showed normal LTM, animals lacking CrebB in the MBs displayed impaired LTM, indicating that the MB is essential for CrebB mediated LTM formation (*Figure 3C*). In contrast, STM was not affected by loss of CrebB in KCs (*Figure 3—figure supplement 1*) and non-induced flies displayed normal LTM (*Figure 3—figure supplement 2A*).

We next examined the effect of CrebB knockout on LTM in each of the three major MB lobe subclasses using lobe specific Gal4 drivers (*c739-Gal4* for α/β, *c305a-Gal4* for α′/β′ and *5-HTR1B-Gal4* for γ neurons) (*Aso et al., 2009*; *Yuan et al., 2005*; *Xie et al., 2013*). We observed a significant reduction in LTM of animals with CrebB knockout in MB α/β neurons compared to control groups (*Figure 3D*). Similarly, flies with a CrebB knockout in MB α′/β′ neurons also exhibited impaired long-term memory performance (*Figure 3E*). However, removing CrebB from MB γ neurons did not affect LTM formation. Performance indices did not differ between the tested groups (*Figure 3F*). Effective knockout of CrebB from MB γ neurons could be confirmed by antibody staining (*Figure 3—figure supplement 3*). LTM performance was also tested in flies with non-induced CrebB knockout. Non-induced control flies of the KC subtype experiments did not show LTM defects (*Figure 3—figure supplement 2B–D*). These findings show that appetitive LTM formation requires CrebB activity in α/β and α′/β′ KCs but not in γ KCs.

## CrebB is required in MBON α3 for LTM formation

For distinct forms of learning, different types of MB intrinsic and extrinsic neurons are required. Appetitive memories require dopaminergic input from the PAM cluster neurons (*Burke et al., 2012*; *Liu et al., 2012*). We used the *GMR58E02-Gal4* line that strongly labels the PAM cluster neurons to mediate CrebB knockout in those DANs. However, flies with *CrebB* null mutant PAM neurons were able to form LTM indistinguishable of the concurrently tested control groups (*Figure 4A*). To verify that CrebB was successfully removed from PAM neurons, antibody staining experiments were conducted. After induction of CrebB knockout in PAM neurons, CrebB::GFP could not be detected anymore in most TH-labeled dopaminergic neurons in the brain region where PAM neurons are located (*Figure 4—figure supplement 1*).

We next assayed if CrebB is needed in MB efferent neurons. Two pairs of cholinergic mushroom body output neurons MBON α3 (also called MB-V3) were reported to be necessary for appetitive LTM (*Plaçais et al., 2013*). The *G0239-Gal4* line is a highly specific driver that expresses *Gal4* exclusively in MBON α3 in the adult brain (*Chiang et al., 2011*). We used this line to express FLP and induce *CrebB* deletion from the genome of MBON α3. Interestingly, memory measured 24 h after appetitive olfactory conditioning was impaired in these flies (*Figure 4B*). Furthermore, LTM performance was not affected under low temperature conditions, in which FLP expression is not induced (*Figure 4—figure supplement 2*). Thus, while dopaminergic input neurons do not require CrebB,

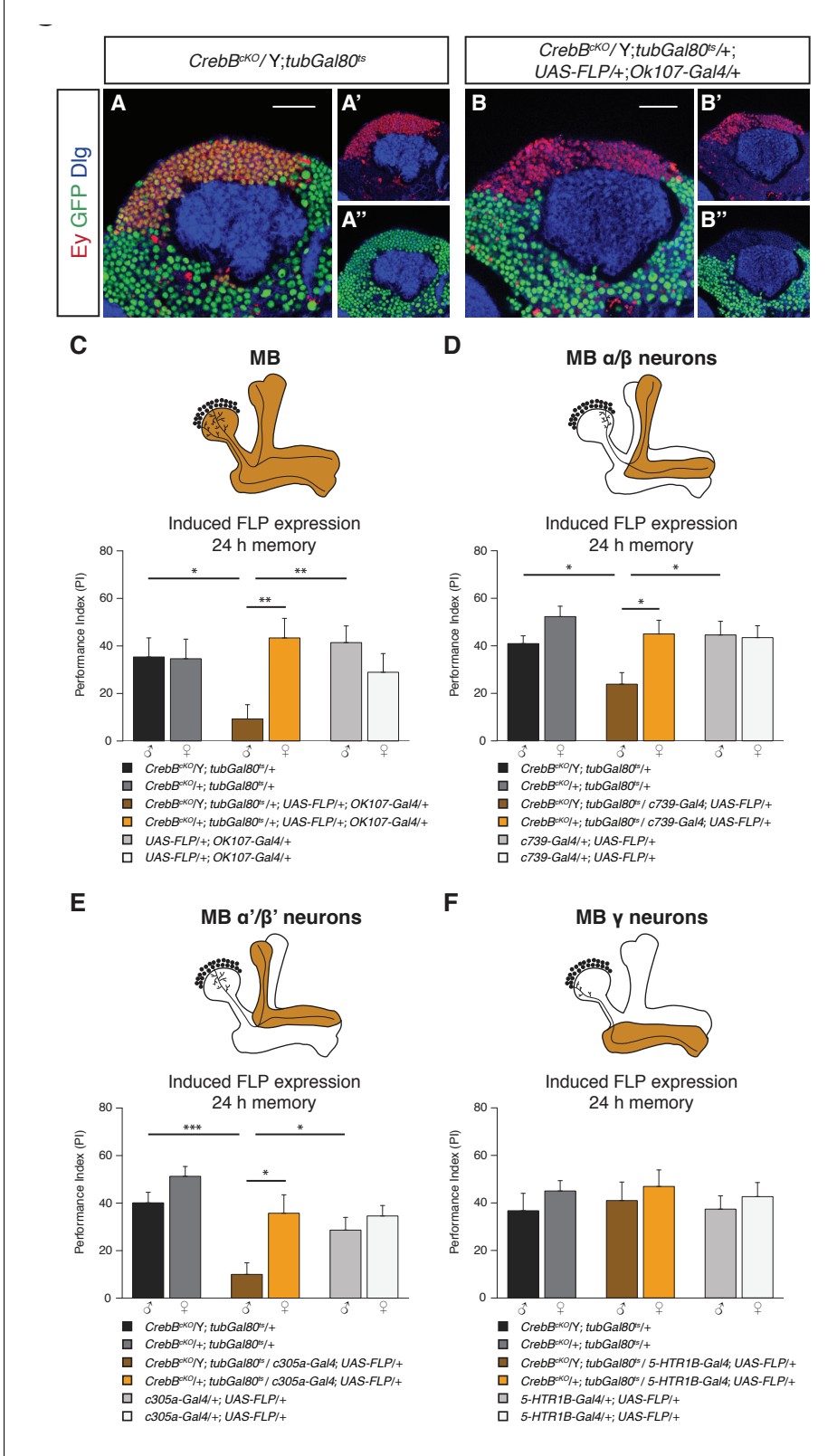

**Figure 3.** CrebB is required in MB α/β and MB α'/β' neurons for LTM formation. (A, B) Brains were stained using anti-GFP (green), anti-Discs large (Dlg, blue) and anti-Eyeless (Ey, red) antibodies, which labels Kenyon cells. (A–A'') CrebB::GFP is expressed in Kenyon cells of *CrebB^cKO; tubGal80^ts* male flies. (B–B'') After induction of mushroom body-specific CrebB::GFP knockout, Ey-expressing Kenyon cells lost GFP expression. Scale bars: 25 μm. (C–F) Different *MB-Gal4* driver lines were used to induce deletion of *CrebB* in Kenyon cells and to test for the requirement of CrebB for 24 h memory. (C) *Figure 3 continued on next page*

*Figure 3 continued*

Induction of CrebB knockout in the entire MB using *OK107-Gal4* severely impaired LTM. N ≥ 10. (**D, E**) The 24 h memory scores were significantly reduced by the knockout of CrebB in MB α/β neurons with *c739-Gal4* (N ≥ 9) and in MB α'/β' neurons with *c305a-Gal4* (N ≥ 8). (**F**) CrebB knockout in MB γ neurons with *5-HTR1B-Gal4* did not impair LTM formation. No significant difference was observed between the tested groups. N ≥ 8. Bar graphs represent the mean and error bars represent the standard error of the mean (SEM). Asterisks denote significant differences between groups; *p<0.05, **p<0.01, ***p<0.001 (Welch two sample *t*-test).

DOI: https://doi.org/10.7554/eLife.39196.007

The following figure supplements are available for figure 3:

**Figure supplement 1.** STM is not affected by CrebB knockout in Kenyon cells.
DOI: https://doi.org/10.7554/eLife.39196.008
**Figure supplement 2.** Non-induced controls display normal 24 h memory.
DOI: https://doi.org/10.7554/eLife.39196.009
**Figure supplement 3.** Successful removal of CrebB::GFP from MB γ neurons.
DOI: https://doi.org/10.7554/eLife.39196.010

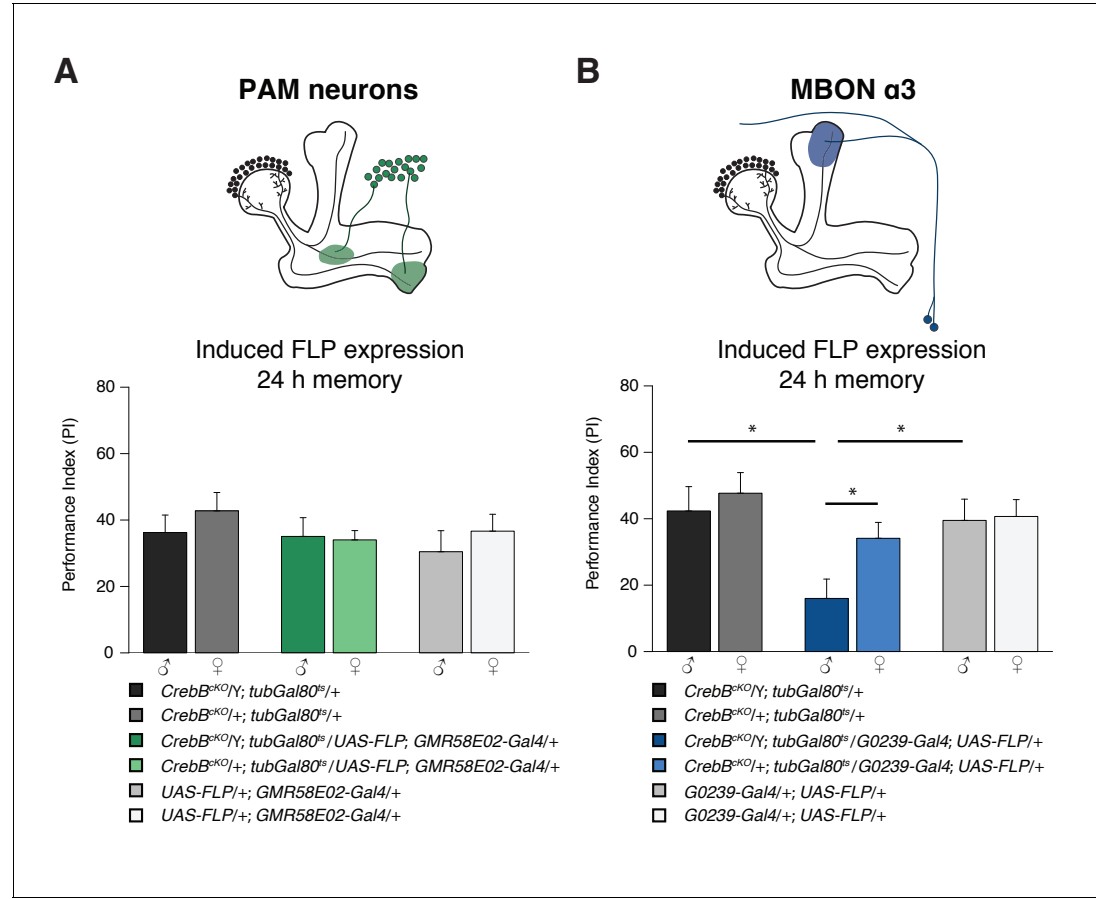

**Figure 4.** The *CrebB* gene is required for LTM in MBON α3. (**A**) Deleting *CrebB* in PAM neurons using the driver *GMR58E02-Gal4* did not affect LTM formation. Performance indices did not differ between the groups. N ≥ 9. (**B**) CrebB knockout in MBON α3 with *G0239-Gal4* impaired LTM. N ≥ 9. Bar graphs represent the mean and error bars represent the standard error of the mean (SEM). Asterisks denote significant differences between groups; *p<0.05 (Welch two sample *t*-test).

DOI: https://doi.org/10.7554/eLife.39196.011

The following figure supplements are available for figure 4:

**Figure supplement 1.** Effective CrebB::GFP knockout in PAM neurons.
DOI: https://doi.org/10.7554/eLife.39196.012
**Figure supplement 2.** Non-induced controls for the CrebB knockout experiment in PAM neurons and MBON α3 show normal LTM.
DOI: https://doi.org/10.7554/eLife.39196.013

MBON α3 requires CrebB for appetitive LTM. This finding supports that LTM is not only encoded within the MB-lobe system, but also involves a memory trace in the MBON extrinsic element.

## Discussion

Formation of long-term memory requires de novo gene expression mediated by CrebB (*Yin et al., 1994*; *Krashes and Waddell, 2008*). Previous studies made use of RNAi-lines to knock-down *CrebB* RNA levels or the expression of a repressor isoform to block CrebB, which resulted in conflicting results. It has been shown that temporal control of inhibiting transgene expression is critical, since expression throughout the development of the flies may produce neuroanatomical damage (*Chen et al., 2012*). Moreover, it has been suggested that defects in LTM correlate with the amount of *CrebB-RNAi* or *CrebB* repressor expression. It seems that the usage of different inducible gene expression systems or the number of transgene copies can strongly influence the outcome (*Hirano et al., 2013*). Experimental parameters, such as when and how strong inhibiting transgenes are expressed may bias the results and affect the study of the role of CrebB in LTM. We therefore performed experiments in which we genetically removed *CrebB$^{cKO}$* specifically in adult flies after completion of brain development. Thus, the removal of the CrebB locus only occurred in post-mitotic, fully- differentiated neurons after eclosion. However, even with adult-specific knockout induction, neuroanatomical abnormalities cannot be completely ruled out.

The cAMP signaling pathway mediates synaptic plasticity required for learning and memory (*Lee, 2015*). The transcription factor CrebB is downstream of the cAMP signaling pathway and its activity regulation during the memory formation process is crucial for LTM (*Yin et al., 1994*; *Yin et al., 1995*; *Fropf et al., 2013*). Consistent with previous studies, we found that pan-neuronal CrebB knockout disrupted LTM, but left STM and MTM intact.

While there is strong consensus that the MB plays a central role in LTM, if protein synthesis and CrebB activity are required in KCs remains debated. A study of *Chen et al., 2012* reported that CrebB mediated gene transcription is required for aversive spaced LTM in DAL neurons but not in MB neurons. In contrast, two groups suggested in recent articles CrebB necessity in the MB for aversive and appetitive LTM (*Hirano et al., 2013*; *Hirano et al., 2016*; *Musso et al., 2017*). Our results, that were obtained using an alternative gene disruption strategy, also demonstrate CrebB necessity in the MBs for LTM formation. Thus, our findings, together with previous studies (*Yu et al., 2006*; *Krashes and Waddell, 2008*; *Hirano et al., 2013*; *Hirano et al., 2016*; *Musso et al., 2017*), convincingly show that CrebB activity is indeed required in KCs for LTM. It is conceivable that in the study of *Chen et al., 2012* CrebB inhibition was insufficient to disrupt LTM formation, leading to conflicting conclusions about the role of CrebB in KCs.

MB α/β, α′/β′ and γ neurons, perform distinct roles in different memory types and phases. Functional studies that used *UAS-shibire$^{ts}$* to block synaptic transmission revealed different temporal requirements of the three major KC subtypes for olfactory associative memory (*Perisse et al., 2013*; *Guven-Ozkan and Davis, 2014*). Output from MB α/β neurons is required for appetitive LTM (*Krashes and Waddell, 2008*; *Trannoy et al., 2011*; *Cervantes-Sandoval et al., 2013*). The α/β lobe neurons are particularly important for LTM, as suggested by multiple other lines of inquiry (*Blum et al., 2009*; *Akalal et al., 2011*; *Huang et al., 2012*; *Ichinose et al., 2015*). It has also been proposed that appetitive LTM depends on the activity of CrebB in MB α/β neurons. Flies expressing a *CrebB* repressor isoform constitutively in MB α/β neurons exhibited a reduced LTM performance (*Krashes and Waddell, 2008*). Coherent with previous work, our results indicate requirement of CrebB in MB α/β neurons for LTM. However, we observed intact LTM after induction of CrebB knockout in MB γ neurons, which have a main role in memory acquisition and expression of STM. It has been reported that expression of an adenylyl cyclase coding *rutabaga* transgene in MB γ neurons could restore STM in *rutabaga* mutant flies, but not LTM (*Schwaerzel et al., 2003*; *Thum et al., 2007*; *Trannoy et al., 2011*). Moreover, output from MB γ neurons was shown to be necessary for STM, though appetitive LTM formation and retrieval did not require γ lobe neuron output (*Trannoy et al., 2011*; *Cervantes-Sandoval et al., 2013*). Nevertheless, a study observed an increased calcium response in MB γ neurons after aversive spaced conditioning that depends on CrebB activity. Expression of a *CrebB* repressor isoform throughout development in MB γ neurons blocked this LTM memory trace and impaired LTM measured 24 h after aversive spaced training (*Akalal et al., 2010*). For appetitive memory, we found unaffected LTM in flies with adult-specific

CrebB knockout in MB γ neurons. Thus, our results suggest that CrebB is not necessary in those neurons for appetitive olfactory LTM.

MB α′/β′ neurons only contribute to around 18% of the total number of KCs and therefore were often neglected (*Aso et al., 2009*). The importance of CrebB to drive LTM formation in α′/β′ lobe neurons has not been assessed before. Our findings argue that LTM formation requires CrebB activity in MB α′/β′ neurons. It has been suggested, that α′/β′ KCs play an essential role in LTM formation and consolidation. Disrupting synaptic activity from MB α′/β′ neurons after appetitive olfactory conditioning prevented LTM, but retrieval of LTM was independent of MB α′/β′ output (*Krashes and Waddell, 2008*; *Cervantes-Sandoval et al., 2013*).

MBONs, which are the downstream neurons of the KCs, can be classified into 21 cell types. Dendrites of different MBON types innervate specific regions of MB lobes and dopaminergic projections align with MBON dendrites, forming distinct compartmental units (*Aso et al., 2014*). Dopamine is released during learning to stimulate plasticity of specific KC-MBON synapses (*Hige et al., 2015*; *Owald et al., 2015*; *Perisse et al., 2016*). DANs from the PAM cluster were shown to be critical for learning the value of carbohydrates (*Burke et al., 2012*; *Liu et al., 2012*). Interestingly, it has been found that MBON α1 synapse onto a subset of PAM neurons that innervates the α1 compartment forming a recurrent network loop, which is necessary for the formation and consolidation of appetitive LTM (*Ichinose et al., 2015*). This feedback circuit motif was also observed in other MB compartments (*Owald et al., 2015*; *Felsenberg et al., 2017*). Recently, plasticity in PAM neurons that lasted not less than 24 h was described. Caloric frustration memory, a new long-lasting memory form, reduced PAM neuron response to glucose (*Musso et al., 2017*). It is possible that modifying efficacy of PAM neuron synapses is involved in appetitive LTM formation. However, we found that LTM does not depend on CrebB activity in MB afferent PAM neurons. This suggests that DANs induce synaptic modifications between KCs and MBONs, but CrebB mediated structural changes in PAM neurons are not required for olfactory appetitive LTM.

Remarkably, we found that flies with CrebB knockout in MBON α3 showed impaired appetitive LTM. This is the first report that LTM requires CrebB activity in MBONs. MBON α3 are efferent from the MB and dendrites were found to project to the MB α lobe (*Pai et al., 2013*; *Plaçais et al., 2013*). It is likely that CrebB mediated transcription in MBON α3 leads to changed efficiency of the postsynaptic sites. The altered KC-MBON synapses would skew the MBON network towards approach behavior (*Owald and Waddell, 2015*). An alternative possibility is that plasticity is induced in MBON α3 presynapses. MBON α3 have presynaptic terminals in the superior medial, the superior intermediate and the superior lateral protocerebrum (*Aso et al., 2014*). A recent report suggested that MBON α3 also connect to DAL neurons, which have axonal processes in the MB and are essential for aversive LTM (*Chen et al., 2012*; *Wu et al., 2017*). Thus, a recurrent loop back to the KCs would be possible. It has been shown that CrebB is required in DAL neurons for aversive spaced LTM, but not for appetitive LTM (*Chen et al., 2012*; *Hirano et al., 2013*). Future studies will be required to reveal the role of this anatomical network for olfactory LTM formation.

While we here specifically focused on the neural circuits requiring CrebB for appetitive olfactory memory it will be interesting to extend the research to aversive memory, since the molecular and neuronal mechanisms are not identical between those two forms of memory. Prior to appetitive learning, it is necessary to deprive flies of food, since motivational drive is critical for memory formation. Furthermore, flies have to be hungry to efficiently express the learned association (*Krashes and Waddell, 2008*; *Colomb et al., 2009*). This imposes limitations and can impede the interpretation of memory performance of distinct memory phases, because different feeding protocols are required.

Our *CrebB^{cKO}* allele may be useful to explore CrebB necessity for other processes, given that the transcription factor CrebB has diverse functions. A wealth of identified *Gal4* driver lines in *Drosophila* provides the possibility to remove CrebB in a precise spatial and temporal manner, and investigate the consequences of CrebB knockout in a large number of cells and contexts.

## Materials and methods

**Key resources table**

*Continued on next page*

*Continued*

| Reagent type (species) or resource | Designation | Source or reference | Identifiers | Additional information |
|---|---|---|---|---|
| Reagent type (species) or resource | Designation | Source or reference | Identifiers | Additional information |
| Genetic reagent (D. melanogaster) | *CrebB^cKO* | this paper | | See Materials and Methods section: Creation of *CrebB^cKO* |
| Genetic reagent (D. melanogaster) | *c739-Gal4* | Hiromu Tanimoto (Tohoku University) | RRID:BDSC_7362 | |
| Genetic reagent (D. melanogaster) | *OK107-Gal4* | Kyoto stock center | RRID:DGGR_106098 | |
| Genetic reagent (D. melanogaster) | *nSyb-Gal4* | Bloomington stock center | RRID:BDSC_51635 | |
| Genetic reagent (D. melanogaster) | *c305a-Gal4* | Bloomington stock center | RRID:BDSC_30829 | |
| Genetic reagent (D. melanogaster) | *5-HTR1B-Gal4* | Bloomington stock center | RRID:BDSC_27636 | |
| Genetic reagent (D. melanogaster) | *GMR58E02-Gal4* | Bloomington stock center | RRID:BDSC_41347 | |
| Genetic reagent (D. melanogaster) | *G0239-Gal4* | Bloomington stock center | RRID:BDSC_12639 | |
| Genetic reagent (D. melanogaster) | *tubGal80^ts* | Bloomington stock center | RRID:BDSC_7019 | |
| Genetic reagent (D. melanogaster) | *nos-Cas9* | Bloomington stock center | RRID:BDSC_54591 | |
| Genetic reagent (D. melanogaster) | *UAS-FLP* (Chr 2) | Bloomington stock center | RRID:BDSC_55806 | |
| Genetic reagent (D. melanogaster) | *UAS-FLP* (Chr 3) | Bloomington stock center | RRID:BDSC_55804 | |
| Antibody | guinea pig anti-CrebB | this paper | | polyclonal anti-CrebB antibody raised against the CrebB full-length protein sequence of isoform F; 1:400 |
| Antibody | rabbit anti-Eyeless | Uwe Waldorf (Saarland University) | | 1:400 |
| Antibody | chicken anti-GFP | Abcam | Cat. #: ab13970 RRID:AB_300798 | 1:1000 |
| Antibody | rabbit anti-GFP | Invitrogen | Cat. #: A-6455 RRID:AB_221570 | 1:1000 |
| Antibody | rabbit anti-Tyrosine hydroxylase | Merck | Cat. #: AB152 RRID:AB_390204 | 1:100 |
| Antibody | mouse anti-Repo | Developmental Studies Hybridoma Bank | Cat. #: 8D12 RRID:AB_528448 | 1:20 |
| Antibody | mouse anti-Discs large | Developmental Studies Hybridoma Bank | Cat. #: 4F3 RRID:AB_528203 | 1:50 |

## Fly strains

Flies (*Drosophila melanogaster*) were generally kept at 25°C and subjected to a 12 h light – 12 h dark cycle. A cornmeal medium supplemented with yeast, fructose and molasses was used to rear the flies. Canton-S was used as wild-type (courtesy of R. Stocker). *c739-Gal4* was obtained from

Hiromu Tanimoto (Tohoku University) and *OK107-Gal4* (106098) was received from Kyoto stock center. *nSyb-Gal4* (51635), *c305a-Gal4* (30829), *5-HTR1B-Gal4* (27636), *GMR58E02-Gal4* (41347), *G0239-Gal4* (12639), *tubGal80^{ts}* (7019), *nos-Cas9* (54591) and *UAS-FLP* (55804, 55806) were obtained from Bloomington stock center.

## Creation of *CrebB^{cKO}*

Genomic DNA of *nos-Cas9* flies was used as template for PCR of the *CrebB* genomic fragments. A 3.3 kb fragment of the *CrebB* encoding region including introns was amplified with primers 'CrebB noStart R1 fw' (gcgaattcGACAACAGCATCGTCGAGGAGAACG) and 'CrebB intr RV re' (gagataTCC TGCCAAGTCGCAACTAAAGGC). The resulting sequence starts with the second codon (Asp GAC) of *CrebB* just after the EcoRI restriction site, and ends within the last intron 45 bp downstream of the facultative exon of isoforms PG, PI, PN and PQ followed by an EcoRV restriction site. For the upstream homology arm a 2.3 kb fragment upstream of the *CrebB* Start codon was amplified with primers 'CrebB CR Not fw' (gggcggcCGCGGAGGTAATGCGGATTTGG) and 'CrebB 5'UTR Spe re' (ttactagtCCTGGCGATCTTCAGCAGCACC). This fragment starts 2312 bp upstream of the *CrebB* start codon within the sequence of the RNA expressing *CR43686* gene, and ends 30 bp upstream of the *CrebB* Start codon within the 5'UTR. For the downstream homology arm a 2.1 kb fragment including the last exon of *CrebB* was amplified with primers 'CrebB intr Xho fw' (ggaccactcgagAA TCGAACTGGAATCGAGGGTCTATC) and 'CrebB 3'UTR Kpn re' (gtggtaCCGTCCCTTCGTCTC TTTTCTACC). This fragment starts within the last intron of *CrebB* 56 bp downstream of the facultative exon of isoforms PG, PI, PN and PQ, and ends within the 3'UTR of the long CrebB isoforms 1131 bp downstream of the Stop codon. All three genomic fragments were subcloned into pBluescript with the appropriate restriction enzymes. After verification of the sequences, the three fragments were assembled in a pBluescript derivative generated in our lab in which we have inserted two FRT sites and the coding sequence of *GFP* into the multiple cloning site allowing us to generate FRT-flanked and GFP-tagged CRISPR templates. The first coding exon of *CrebB* lacking the *CrebB* Start codon was fused in frame with *GFP*. One of the FRT sites is located immediately upstream of the *GFP* coding sequence between the 2.3 kb upstream homology arm and *GFP*, the other FRT site is located between the 3.3 kb *CrebB* coding region and the 2.1 kb downstream homology arm. Thus, upon FRT mediated recombination the GFP-tagged *CrebB* coding region from the first coding exon to the second last coding exon will be removed.

For the first guide RNA oligos 'CRISPR CrebB Start sense' (cttcGATCGCCAGGATCGGCAACA) and 'CRISPR CrebB Start antisense' (aaacTGTTGCCGATCCTGGCGATC) were annealed and ligated into BbsI digested pU6-BbsI-chiRNA vector. This guide RNA will target Cas9 to cut 2 bp upstream of the *CrebB* Start codon. For the second guide RNA oligos 'CRISPR CrebB intr sense' (cttcGGAC-CACTCGTAAATCGAAC) and 'CRISPR CrebB intr antisense' (aaacGTTCGATTTACGAGTGGTCC) were annealed and ligated into BbsI digested pU6-BbsI-chiRNA vector. This guide RNA will target Cas9 to cut 61 bp downstream of the facultative exon of isoforms PG, PI, PN and PQ. The CRISPR sites are missing in the template DNA, where they were replaced with the FRT sites. A mix containing 0.4 µg/µl template and 0.2 µg/µl of each guide RNA plasmid was injected into *nos-Cas9* expressing flies. The injected flies were crossed with *w^{1118}* mutants and their offspring screened for GFP expressing larvae. GFP positive flies were crossed with *FM7* balancer flies to establish a stable line. The presence of the GFP-tag and the FRT sites was verified by PCR and sequencing of the genomic DNA of the homozygous *CrebB^{cKO}* stock.

## Anti-CrebB antibody production

*CrebB* cDNA was PCR amplified from EST clone RT01009 and cloned into the pGEX-6P-1 plasmid to produce a Gst-tagged version in bacteria. The resulting *Gst-CrebB* coding sequence had a small in-frame deletion removing some Glycine residues of the N-terminal Glycine stretch but the rest of the sequence was unaltered. Bacterially expressed Gst-CrebB was purified with Glutathione Sepharose beads (GE healthcare) and injected into guinea pigs for antibody production (eurogentec).

## Antibody staining

Flies, in which CrebB knockout was induced, were moved for 6 days to 29°C before antibody staining experiments. Male adult brains were dissected in phosphate-buffered-saline (PBS) and fixed at

room temperature for 30 min with a 3.7% formaldehyde solution (in PBS). Brains were washed at least five times with PBST (PBS with 0.3% triton X-100) before primary antibodies were added for overnight incubation at 4°C. The following primary antibodies were used: rabbit anti-Eyeless (1:400, courtesy of Uwe Waldorf), chicken anti-GFP (1:1000, Abcam ab13970), rabbit anti-GFP (1:1000, Invitrogen A-6455), guinea pig anti-CrebB (1:400), rabbit anti-Tyrosine hydroxylase (1:100, Merck AB152), mouse anti-Repo (1: 20, Developmental Studies Hybridoma Bank 8D12), mouse anti-Discs large (1:50, Developmental Studies Hybridoma Bank 4F3). Brains were washed again before the overnight incubation at 4°C with secondary antibodies. The used secondary antibodies were conjugated with Alexa fluorescent proteins (488, 567 or 647; Molecular Probes) and diluted 1:200. After a last washing step, brains were mounted in Vectashield H-1000 or Vectashield with DAPI H-1200 (Vector Laboratories) on a microscope slide. Samples were imaged with Leica SP5 confocal microscope and the images were processed with Imaris Bitplane 9.2 and Adobe Photoshop CS6.

## Appetitive olfactory conditioning experiments

Memory experiments were performed at 23–25°C and 70–75% relative humidity. Conditioning was carried out in dim red light and tests were done in darkness. The used conditioning apparatus is based on *Tully and Quinn, 1985* and was modified to perform four experiments in parallel (*Schwaerzel et al., 2002*); CON-Elektronik, Greussenheim, Germany). The odors benzaldehyde (Fluka, 12010) and limonene (Sigma-Aldrich, 183164) were used. 60 µl of benzaldehyde was applied in plastic containers measuring 5 mm in diameter and 85 µl of limonene was applied in plastic containers measuring 7 mm in diameter. Odor delivery was effected with a vacuum pump adjusted to a flow rate of 7 l/min. Filter papers were soaked with distilled water or with a 1.5 M sucrose (Sigma-Aldrich, 84100) solution the day before the experiment and left to dry at room temperature overnight.

19–21 h before conditioning groups of 60–100 flies were put into plastic vials with wet cotton wool on the bottom for starvation. For appetitive olfactory conditioning, flies were loaded in tubes lined with water filter papers. After an acclimatization period of 90 s, a first odor was presented for 2 min. Then, the odor was removed and animals were transferred within 60 s to tubes lined with sucrose filter papers. Subsequently, the second odor was presented for 2 min.

For the memory tests, flies were moved to a two-arm choice point where they could choose between limonene and benzaldehyde for 2 min. After this period, the number of flies within each arm was counted and a preference index was calculated.

$$PREF = ((N_{paired\ odor} - N_{control\ odor})\ 100)/N_{total}$$

One memory experiment consisted of two groups with reciprocal conditioning, in which the sucrose paired odor was exchanged. The preference indices from these two groups were averaged to calculate a performance index (PI).

To measure short-term memory, flies were tested immediately after conditioning. Middle-term memory was examined 3 h after conditioning. For those experiments, animals were put back into starvation vials after training and were starved until the test. Flies tested for 24 h memory were put in food vials after conditioning for 3–5 h and then transferred to starvation vials until the test.

Flies used for memory experiments were reared at 18°C and collected after hatching (0–3 d old flies). To induce expression of flippase, flies were moved for 6 d to 29°C. For the starvation period before conditioning and until the test, those animals were kept at 25°C. Flies used for the non-induced FLP expression experiments were kept at 18°C after collection (for 6 days) and during the starvation time. Memory experiments with those flies were performed at 20–22°C.

## Data analysis

To compare PIs between two groups the Welch two sample *t*-test was used. Statistical analyses and graphical representation of the data were performed using R version 3.4.1.

## Acknowledgements

We would like to thank H Tanimoto, Kyoto stock center and Bloomington stock center for fly strains. We also thank U Waldorf for providing anti-Eyeless antibody. We would like to thank Unifr Bioimage Facility and J Bernardo-Garcia for their support and colleagues of the Sprecher lab for valuable discussions. This work is part of the SynaptiX RTD funded by SystemsXch.

## Additional information

### Funding

| Funder | Grant reference number | Author |
|---|---|---|
| Bundesbehörden der Schweizerischen Eidgenossenschaft | SynaptiX | Simon G Sprecher |
| Novartis Stiftung für Medizinisch-Biologische Forschung | 18A017 | Simon G Sprecher |
| Schweizerischer Nationalfonds zur Förderung der Wissenschaftlichen Forschung | CRSII5_180316 | Simon G Sprecher |

The funders had no role in study design, data collection and interpretation, or the decision to submit the work for publication.

### Author contributions

Yves F Widmer, Conceptualization, Data curation, Formal analysis, Investigation, Methodology; Cornelia Fritsch, Investigation, Writing—original draft; Magali M Jungo, Silvia Almeida, Investigation; Boris Egger, Investigation, Visualization; Simon G Sprecher, Conceptualization, Supervision, Funding acquisition, Writing—original draft, Project administration

### Author ORCIDs

Yves F Widmer (iD) http://orcid.org/0000-0002-8880-1392
Simon G Sprecher (iD) http://orcid.org/0000-0001-9060-3750

### Decision letter and Author response

Decision letter https://doi.org/10.7554/eLife.39196.016
Author response https://doi.org/10.7554/eLife.39196.017

## Additional files

### Supplementary files

• Transparent reporting form
DOI: https://doi.org/10.7554/eLife.39196.014

### Data availability

All data is included in the manuscript.

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
