## [Decision Letter]

Thank you for submitting your article "Multiple neurons encode CrebB dependent appetitive long-term memory in the mushroom body circuit" for consideration by *eLife*. Your article has been reviewed by three peer reviewers, including Leslie C Griffith as the Reviewing Editor and Reviewer #1, and the evaluation has been overseen by K VijayRaghavan as the Senior Editor. The following individuals involved in review of your submission have agreed to reveal their identity: Alex C Keene (Reviewer #2).

The reviewers have discussed the reviews with one another and the Reviewing Editor has drafted this decision to help you prepare a revised submission.

Summary:

This paper brings conditional CRISPR tools to the problem of localization of CrebB function in memory formation. The authors have generated and validated a new tool for the investigation of the role of CREB that will likely be used by many investigators. They march through several of the cell types required for appetitive LTM formation and show that some, but not all, require an intact CrebB gene.

Essential revisions:

There are several critical controls that need to be done in order for this paper to be a convincing contribution.

1) The genotypes in the figures suggest that *Gal80^ts^* was not used for the anatomical experiment, but rather that *FLP* was activated developmentally for the anatomy. It is problematic to use the developmentally expressed FLP and loss of GFP signal to interpret the behavioral data, which is supposed to restrict FLP expression to the adult. Authors must show anatomical data for animals that had adult-specific FLP induction. Lack of phenotype for some of the *Gal4*s could be due to lack of adequate conversion.

2) This raises a second point. The anatomical figures are of very low quality. Higher mag images and a counterstain for the areas where FLP is expressed are required to allow assessment of the efficacy of the adult-specific FLP outs.

3) The authors also need to do temperature controls. They should show how experimental flies behave (STM and LTM) when kept at the low temperature through development and adulthood. This is important for the conclusion that the manipulation of CrebB was restricted to the adult fly and that this changed LTM.

---

## [Author Response]

There are several critical controls that need to be done in order for this paper to be a convincing contribution.1) The genotypes in the figures suggest that Gal80^ts^ was not used for the anatomical experiment, but rather that FLP was activated developmentally for the anatomy. It is problematic to use the developmentally expressed FLP and loss of GFP signal to interpret the behavioral data, which is supposed to restrict FLP expression to the adult. Authors must show anatomical data for animals that had adult-specific FLP induction. Lack of phenotype for some of the Gal4s could be due to lack of adequate conversion.

This is indeed a valid point. We have now performed antibody staining with flies that were treated identical with flies used for memory experiments. We replaced the panels in the corresponding figures accordingly.

In addition, we conducted antibody staining with genotypes that did not show a LTM phenotype after *CrebB* deletion. CrebB::GFP removal from MB g neurons (*5-HTR1B-Gal4*) and PAM neurons (*GMR58E02-Gal4*) were confirmed and the results are shown in figure supplements (Figure 3—figure supplement 3 and Figure 4—figure supplement 1).

2) This raises a second point. The anatomical figures are of very low quality. Higher mag images and a counterstain for the areas where FLP is expressed are required to allow assessment of the efficacy of the adult-specific FLP outs.

We apologize for the quality of previous figures and have now performed a set of experiments to improve anatomical figures displayed at better resolution. The co-localization between CrebB and GFP (Figure 1 and Figure 1—figure supplement 1) and the FLP mediated CrebB::GFP knockout (Figure 2 and Figure 3) can be better assessed in the new images.

3) The authors also need to do temperature controls. They should show how experimental flies behave (STM and LTM) when kept at the low temperature through development and adulthood. This is important for the conclusion that the manipulation of CrebB was restricted to the adult fly and that this changed LTM.

It is indeed true that testing memory performance of flies with non-induced FLP expression is an important control. We have now conducted non-induced FLP expression control experiments, in which flies were kept continuously at 18°C, for all the used genotypes. The data is displayed in figure supplements.